# FDTD-Based Electromagnetic Modeling of Dielectric Materials with Fractional Dispersive Response

Luciano Mescia [1],* , Pietro Bia [2] and Diego Caratelli [3,4]

1 Department of Electrical and Information Engineering, Politecnico di Bari, 70125 Bari, Italy
2 Design Solution Department, Elettronica SpA, 00131 Rome, Italy; pietro.bia@elt.it
3 Department of Electrical Engineering, Eindhoven University of Technology, P.O. Box 513, 5600 MB Eindhoven, The Netherlands; diego.caratelli@antennacompany.com
4 Department of Research and Development, The Antenna Company, High Tech Campus, 5656 AE Eindhoven, The Netherlands
* Correspondence: luciano.mescia@poliba.it; Tel.: +39-080-5963808

**Abstract:** The use of fractional derivatives and integrals has been steadily increasing thanks to their ability to capture effects and describe several natural phenomena in a better and systematic manner. Considering that the study of fractional calculus theory opens the mind to new branches of thought, in this paper, we illustrate that such concepts can be successfully implemented in electromagnetic theory, leading to the generalizations of the Maxwell's equations. We give a brief review of the fractional vector calculus including the generalization of fractional gradient, divergence, curl, and Laplacian operators, as well as the Green, Stokes, Gauss, and Helmholtz theorems. Then, we review the physical and mathematical aspects of dielectric relaxation processes exhibiting non-exponential decay in time, focusing the attention on the time-harmonic relative permittivity function based on a general fractional polynomial series approximation. The different topics pertaining to the incorporation of the power-law dielectric response in the FDTD algorithm are explained, too. In particular, we discuss in detail a home-made fractional calculus-based FDTD scheme, also considering key issues concerning the bounding of the computational domain and the numerical stability. Finally, some examples involving different dispersive dielectrics are presented with the aim to demonstrate the usefulness and reliability of the developed FDTD scheme.

**Keywords:** fractional differential equations; Riemann–Liouville derivative; dielectric relaxation; dispersive media; finite difference time domain; Maxwell's equations

## 1. Introduction

Fractional calculus (FC) was introduced more than 300 years ago as a generalization of classical derivative and integral definitions. It is receiving increasing attention for a growing number of applications in different sciences such as physics, biology, chemistry, engineering, finance, mechanics, optics and, in particular, for modeling physical phenomena related to non-Markovian processes, signal and image processing, dielectric relaxation, viscoelasticity, electromagnetism, control theory, pharmacokinetics, fluids, heat transfer, and so on [1–7].

It is well known that the description of real-world problems using mathematical models based on the differentiation and integration provides a huge contribution to science. However, the classical PDE models fall short of describing many complex anomalous systems and phenomena featuring persistent memory effects and long-range interaction in the medium. The integer-order differential and integral operators are local, i.e., the interactions between two domains happen only through the contact. On the other hand, the space and time-fractional integro-differential operators are not local. This means that interactions among components extend to a neighborhood of each component (space non-locality) as well as being able to model systems in which the reaction to an external excitation is not instantaneous but depends on the history of the system (time non-locality).

For this reason, the integral and differential operators of non-integer order are more effective tools for describing media and systems with non-local and hereditary properties of power-law type, long-range memory and/or fractal properties. Moreover, the unusual properties of the fractional derivatives and integrals, including the violation of the standard product, chain, and semigroup rules, can be used to describe unusual properties of complex systems and media [8].

Although the concept of derivative of non-integer order can be used to model complex natural processes, its physical meaning can still be grasped in a fairly easy way. The application of FC opens the door to entirely new branches of thought. In this paper, we show how the fractional derivative operators can be useful tools in electromagnetism. Initially, we provide a brief overview of the application of FC to dielectric relaxation modeling. Then, we illustrate a generalization of Maxwell equations using FC theory to study transient wave propagation in arbitrary dispersive media. The different topics and challenges regarding the incorporation of a general fractional polynomial series in a home-made FC-based FDTD algorithm are discussed in detail. The key issues concerning the numerical dispersion and stability of the FDTD scheme are highlighted using the von Neumann spectral approach and the zeros of the characteristic polynomial equation as a function of the Courant factor. The design and implementation of dedicated UPML absorbing boundary condition to terminate the outer boundary of the computational domain are also illustrated in combination with the time-marching scheme. The implementation of the TFSF procedure is discussed, too. Examples illustrating the electromagnetic field propagation in different dispersive dielectrics are presented with the aim of demonstrating the capability of the developed methodology to resolve challenging electromagnetic problems.

## 2. Fractional Calculus in Electromagnetic Theory

### 2.1. Fractional Vector Calculus

As it is well known, Maxwell's equations are the theoretical background for understanding the classical electromagnetic phenomena. In their differential form, Maxwell's equations involve integer-order calculus applied to vector differential operators of the first order. Therefore, the extension of the fractional paradigm in electromagnetism needs a consistent FVC theory. During the last few decades, different fractional generalizations of vector operators such as gradient, divergence, curl, and Laplacian have been suggested. The potential applications of fractional calculus in electromagnetic theory were explored in [9,10] with focus on multipole expansion, theory of images in electrostatics, scalar Helmholtz equation. The definition of the FO curl operator in electromagnetic theory was introduced in [11] to derive solutions for Maxwell's equations by fractionalizing the principle of duality. This approach was applied to solve some reflection and diffraction problems [12]. The calculation of the source distributions corresponding to fractional dual solutions and intermediate fractional dual solutions was illustrated in [13]. The behavior of fractional dual solutions in metamaterials with negative permittivity and permeability, as well as homogeneous and non-homogeneous chiral media was investigated in [14–16] using a higher-order fractional curl operator. The definition of FO gradient and FO divergence were proposed in 1998 [17] and 2006 [18], respectively, as well as a consistent formulation of FVC based on Riemann–Liouville integration and the Caputo differentiation was suggested in [19]. Using this approach, the fractional differential and integral vector operators were defined and the fractional Green, Stokes, and Gauss theorems were proved. After 2008, other articles illustrating special aspects of FVC began to appear in the scientific literature. Using Grünwald–Letnikov fractional calculus, the classical vectorial operators of gradient, divergence, curl, and Laplacian could be generalized to fractional orders and an extension of the Helmholtz decomposition theorem for both fractional time and space was proposed [20]. Based on these results, the fractional versions of the classical Green, Stokes, and Ostrogradski–Gauss theorems were introduced in [21]. Moreover, higher-order fractional Green and Gauss formulas were derived in [22]. However, just in 2021, a gen-

eralization of FVC was proposed to include a general form of non-locality in kernels of fractional vector differential and integral operators [23].

Thanks to the aforementioned developments in FVC, different approaches have been suggested to fractionalize the electromagnetic field equations. The idea of such a generalization relies on the application of material constitutive relations extended in terms of FO derivatives. Using the fractional differential forms, the non-local Maxwell's equations were introduced in [24]. The authors (i) defined fractional vector and scalar potentials, (ii) developed fractional conservative law for the electric charge, (iii) derived fractional wave equations and fractional Poynting theorem. A solution to the source-free fractional wave equation in isotropic and homogeneous dielectric media was presented in [25] with the aim of maintaining the physical units of the system for any value of the fractional operator exponent. Moreover, it was highlighted that the resulting fractional space–time waves are independent of the physical structure, chemical composition, and polarization of the material. The fractional wave equation in space-time and in a conducting media was introduced and analyzed in [26]. The authors showed that fractional electromagnetic fields have different characteristics compared to the classical ones and they exhibit anomalous behavior that is known in the literature as centrovelocity. Solutions for fractional generalization of the Laplacian were derived in [27] with the aim of describing the electric fields in non-local media with power-law spatial dispersion. In particular, a generalizations of Debye's permittivity and a weak spatial dispersion of power-law type in plasma-like media were suggested. Using the formulation of TF electrodynamics with the Riemann–Silberstein vector, a compact form of FO Maxwell equations was presented in [28]. The proposed formulation allows for inclusion of energy dissipation and establishes a relation between the TF Schrödinger equation and TF electrodynamics. Using the two-sided Ortigueira–Machado derivative, the plane wave signal propagation in media described by FO model was analyzed in terms of system causality [29]. In particular, causality of the transfer function is proven for certain values of the asymmetry parameter, corresponding to the left-sided Grünwald–Letnikov derivative. In addition, an exact solution of the spherical and cylindrical wave equations in fractional dimensional space was presented in [30,31].

### 2.2. Fractal Media

The modeling of electromagnetic wave propagation, radiation, and scattering in complex fractal structures has attracted growing attention and seen an increasing number of applications. Moreover, fractal models are becoming popular as they allow describing media of great complexity and rich structure using just a few parameters. Starting from the idea that isotropic and anisotropic fractal media in an Euclidean space can be replaced by some fractional continuous mathematical model [32,33], the electromagnetic wave propagation in fractal media can be studied through an ordinary continuum model with non-integer dimensional space. Using the concepts of power-law density of states and fractional-order integration, fractal electrodynamics of fractal distribution of charges, currents, and fields was described in [4,34]. Electromagnetic equations in fractional space were presented in [35,36]. In particular, a modified vector differential operator in a D-dimensional fractional space was defined with the aim of introducing a fractional space generalization of differential electromagnetic equations in a far-field region. In [37], the wave propagation in fractal media was studied by solving a generalized vector Helmholtz equation in terms of plane wave solutions. In [38], the authors derived electromagnetic equations in generally anisotropic fractal media using a dimensional regularization approach. Such an approach allows to define fractal gradient, divergence, and curl operators, satisfying the four basic identities of vector calculus as well as the Helmholtz decomposition theorem. Furthermore, it readily leads to the generalization of Green–Gauss and Stokes theorems, as well as of the charge conservation equation in anisotropic fractals. Again, the study of transmission and reflection of electromagnetic waves at dielectric–fractal interfaces was presented in [39]. The authors assumed that the fractionality exists only along the z-axis and that the permeability of the fractional medium is approximately the same as in the integer space. Therefore,

they derived the expressions for transmission and reflection coefficients for parallel and perpendicular polarizations, as well as the general solution for plane waves.

### 2.3. Dielectric Media

Dielectric materials have an important role in science and technology. Hence, their relaxation properties are the subject of numerous new research activities. For instance, the dielectric response knowledge of an unknown material inside a lossy and dispersive structures is of paramount importance in GPR techniques and material characterization [40–42]. The accurate knowledge of frequency-dependent dielectric susceptibility is also relevant in microwave computational dosimetry, dielectric spectroscopy and imaging for early-stage cancer diagnostics and treatment, as well as in the study of the materials involved in the production and processing of crops and food of agricultural origin [43,44]. Furthermore, dielectric properties play an important role in the analysis of body area networks, nano-networks in living biological tissues, and for in-body electromagnetic communications [45–48].

Great efforts have been made to model the frequency dispersive behavior of the dielectric susceptibility for a variety of materials in different frequency ranges. Simple models based on exponential decay in time of the relaxation function were introduced in [49–51]. Numerous experimental studies have demonstrated that anomalous relaxation and diffusion processes can exhibit a time behavior that is different from the exponential law. These dynamic dielectric properties seem to be a feature of disordered media as spin or dipole glasses, polymers, biopolymers, emulsions, and microemulsions, disordered ferroelectrics, biological cells and tissues, porous materials [52]. Thanks to the relevant advances in measurement science, a significant amount of experimental data has been collected to validate the existence of non-exponential relaxation processes. Meanwhile, the attention of researchers has been focused on modeling dielectric response over broader frequency ranges [51–53]. The resulting models allow evaluating performance indexes as well as optimizing the dielectric response so as to meet given design specifications or to generate unusual electromagnetic properties [54,55]. The right knowledge of the dielectric response over broad frequency ranges is of utmost importance for the development of accurate theoretical models and computational techniques aimed at determining the electromagnetic field propagation properties inside complex and disordered dielectric materials. In particular, numerical simulations could be very useful for identifying the main parameters involved in non-invasive diagnosis and medical sensors, to evaluate the characteristics of in-body communications among nano devices in terms of data rate and transmission range [48], as well as to provide guidelines for computational dosimetry and specific therapeutic approaches.

Important empirical relaxation laws or relationships to describe non-exponential relaxation processes causing broadness, asymmetry, and dielectric dispersion excess were introduced in [56–63]. Even though these relationships have been moderately successful in describing relaxation phenomena in different condensed matter systems, they do not entirely clarify the anomalies in the dielectric response of disordered systems from the physical point of view. With the help of a model based on a self-similar relaxation process, in [64], the authors attempted to understand the physical process leading to the appearance of non-exponential relaxation in the Cole–Davidson expression as well as the meaning of the non-integer parameter which appears in it. They stressed that contrary to the TRD concept, the self-similar one results in a relaxation with discontinuous behavior, such that the times when the interaction with an external field exists are distributed over some fractal set. In [65], the authors derived different relaxation functions assuming relaxation times bounded between the upper and lower limits of self-similarity. Moreover, they predicted that at times shorter than that corresponding to the lowest self-similarity level, the relaxation should be of Debye-like type, whatever the pattern of non-classical relaxation is at longer times. In [66], the author attempted to provide a unified interpretation of general relaxation phenomena by exploring the physical mechanism underlying dielectric

relaxation phenomena. In particular, it was demonstrated that dielectric materials possess the normal structure and the nematic phase in which the slowly fluctuating dielectric response is influenced and constrained by the collective one.

Even if there is no universal law capable of describing any dielectric response in any frequency band, the proposed empirical relationships share the common feature of power law decay for large times. This behavior allows deriving integro-differential equations with time derivatives and integrals of non-integer order that can be incorporated into Maxwell's equations. The use of fractional derivative techniques to describe the anomalous dielectric response, in the time domain, of an inhomogeneous medium is demonstrated in [65]. Here, the authors established the relationship between anomalous relaxation and dimensionality of a temporal fractal ensemble characterizing the non-equilibrium state of a medium. An equation for the relaxation function containing FO integral and differential operators was derived and solved in [64]. A model based on viscoelastic analysis techniques employing FO operators was proposed in [67] to evaluate the dielectric response of a material in both radio frequency and terahertz bandwidths. Moreover, a survey on the main dielectric models and their characterization in terms of FO differential operators can also be found in [7]. In [68], it was proved that electromagnetic propagation in a wide class of dielectric media can be described by FO differential equations in the time domain. On the basis of recent non-local effects observed in EMMs, a fractional Drude model was introduced to describe electromagnetic characteristics of metamaterials [69]. A scientific discussion on the analysis of perfect cloaking based on FO formulation of electromagnetics was initiated in [70]. In particular, a two-dimensional cloak was numerically demonstrated in media described by FO models. Moreover, a tuning of the signal arrival time was obtained by means of a small perturbation of the time-derivative orders in Maxwell's equations [71].

Other key issues which should not be neglected pertain to the development and implementation of numerical methods for finding approximate solutions of FO Maxwell's equations, especially when fractional problems involving large spaces and long-time iterations need to be solved. In fact, due to the non-locality of the fractional operators, the computational cost concerning the numerical implementation of fractional differential equations is much heavier than that relevant to classical integer-order equations. For example, the relative complex permittivity functions corresponding to non-integer powers of $j\omega$ require special treatments aimed at embedding the approximation of fractional derivatives inside the kernel of the FDTD algorithm [72,73]. These problems represent a real challenge from a computational standpoint. Therefore, it is extremely useful to devise and develop fast, efficient, and reliable numerical methods for solving FO electromagnetic problems.

## 3. Fractional Dielectric Response Models

It is common knowledge that dispersive media exhibit frequency-dependent permittivity and conductivity functions when exposed to time-harmonic electromagnetic fields. They are often encountered in nature, such as in biological tissues and cells, polymers, biopolymers, emulsions and microemulsions, rocks, soils, ice, snow, and plasma. As a result, the modeling of dispersive materials is important for studying several electromagnetic phenomena. Moreover, the permittivity spectrum provides a unique dielectric signature very useful for sensing purposes.

Polarization can be regarded as an ordering process in space which electrical charges undergo when they are subject to an external electric field. In particular, under the influence of a step-wise electric field, the buildup of material polarization takes a finite time interval before approaching its maximum value. In causal linear and isotropic medium, the electric induction $\boldsymbol{\mathcal{D}}$ is related to the electric field $\boldsymbol{\mathcal{E}}$ by the integral relation

$$\boldsymbol{\mathcal{D}}(t) = \varepsilon_0 \boldsymbol{\mathcal{E}}(t) + \varepsilon_0 \int_{-\infty}^{t} \chi_e(\tau) \boldsymbol{\mathcal{E}}(t - \tau) \mathrm{d}\tau \tag{1}$$

or in time-harmonic form

$$\mathbf{D}(\omega) = \varepsilon_0[1 + \chi_e(\omega)]\mathbf{E}(\omega) = \varepsilon(\omega)\mathbf{E}(\omega) \tag{2}$$

as well as

$$\mathbf{D}(\omega) = \varepsilon_0\mathbf{E}(\omega) + \mathbf{P}(\omega) \tag{3}$$

where $\varepsilon_0$ and $\chi_e$ are the vacuum permittivity and electric susceptibility, respectively, $\varepsilon(\omega)$ is the complex permittivity, $\omega$—the angular frequency. Moreover, the polarization vector $\mathbf{P}(\omega) = \varepsilon_0\chi_e(\omega)\mathbf{E}(\omega)$ accounts for the local displacement of bound charge in the material. Moreover, the dielectric relaxation phenomenon can be expressed in terms of time-domain response function $\phi(t)$ through the relation

$$\phi(t) = \mathcal{F}^{-1}\left\{\frac{\varepsilon(\omega) - \varepsilon_\infty}{\varepsilon_s - \varepsilon_\infty}\right\} \tag{4}$$

where $\mathcal{F}^{-1}$ is the inverse Fourier transform, $\varepsilon_s$ and $\varepsilon_\infty$ are the permittivities under static voltage and at infinity frequency, respectively. The classical Debye expression has been widely used to model the dielectric response of simple materials at specific frequency bandwidths [49]. The corresponding time-harmonic response function is

$$\phi(\omega) = \frac{1}{1 + j\omega\tau} \tag{5}$$

where $\tau$ is the relaxation time and $j = \sqrt{-1}$ is the imaginary number. In this case, the time-domain response has a purely exponential behavior

$$\phi(t) = \phi(0)\exp(-t/\tau) \tag{6}$$

and it solves the simple differential equation

$$\frac{\mathrm{d}\phi(t)}{\mathrm{d}t} + \frac{1}{\tau}\phi(t) = 0 \tag{7}$$

However, even if the Debye model is only viable for non-interacting randomly oriented dipoles freely floating in a neutral viscous liquid, it has been often used as the starting point for investigating relaxation responses of dielectrics.

Many experimental investigations of complex systems have demonstrated that the relaxation behavior strongly deviates from the exponential Debye pattern exhibiting a broad distribution of relaxation times. From the macroscopic point of view, the interaction of the electromagnetic energy with heterogeneous mixtures of dielectric materials is given by the different dynamic processes involving reorientation of bipolar molecules, interfacial or space-charge polarization, ionic diffusion, motion of the molecules, relaxations due to the non-spherical shape, resonant phenomena pertaining to molecular, atomic, and electronic vibrations [52,53]. The ionic, interfacial, and dipolar polarizations cause a frequency dispersion pattern of permittivity with drops occurring at relaxation frequency marked by a loss in electromagnetic energy [74]. This anomalous behavior has motivated the development of empirical modification of the Debye relationship. From a mathematical point of view, the anomalies in a dielectric response can be explained by observing that the disordered nature and microstructure of the systems yield multiple relaxation times which result in a generally non-symmetric response in the time domain [75,76]. Therefore, as it appears in (5), multiple Debye terms can be used to accurately match experimental behavior, especially when a wide frequency range has to be considered. To overcome this

drawback while enabling a more accurate modeling as compared to the simple Debye-based formulation, the following fractional power laws were proposed: the CC function [56]

$$\phi(\omega) = \frac{1}{1 + (j\omega\tau)^\alpha} \tag{8}$$

the CD function [57]

$$\phi(\omega) = \frac{1}{(1 + j\omega\tau)^\beta} \tag{9}$$

the HN function [58]

$$\phi(\omega) = \frac{1}{[1 + (j\omega\tau)^\alpha]^\beta} \tag{10}$$

and the Raicu function [62]

$$\phi(\omega) = \frac{1}{[(j\omega\tau)^\gamma + (j\omega\tau)^\alpha]^\beta} \tag{11}$$

where the free parameters $0 \leq \alpha, \beta, \gamma \leq 1$ account for the shape of the relaxation behaviors. However, taking the inverse Fourier transform of Equations (8)–(11), a time-domain response function strongly deviating from the exponential one can be inferred.

There also exist materials with heterogeneous, inhomogeneous, and disordered structure at both microscopic and mesoscopic scales whose relaxation data cannot be interpreted in an accurate way using the empirical relationships (8)–(11). In view of this problem and to enable an extended fitting capability of the experimental data over a broad frequency range, other models have been discussed in the literature [7,77]. A modified Havriliak–Negami relationship has recently recently proposed with the aim of fitting experimental data exhibiting a two-power-law relaxation pattern [63]. A dielectric relaxation model based on Debye functions with relaxation times following a modified Weibull distribution was illustrated in [78]. In the same paper, the authors derived a new closed-form expressions for the real and imaginary parts of complex permittivities in terms of generalized hypergeometric functions. Moreover, in [79], the authors extended the conventional dielectric relaxation analysis by including stochastic resetting dynamic.

Using a general fractional polynomial series approximation, we developed the following time-harmonic relative permittivity function

$$\varepsilon_r(\omega) = \frac{\varepsilon(\omega)}{\varepsilon_0} = \frac{a_0 + \sum_{k=1}^{N} a_k (j\omega)^{\alpha_k}}{b_0 + \sum_{l=1}^{M} b_l (j\omega)^{\beta_l}} = \frac{\mathcal{P}(j\omega)}{\mathcal{Q}(j\omega)} \tag{12}$$

where $a_k, b_k, \alpha_k, \beta_k$ denote suitable real-valued parameters. However, the different terms at both numerator and denominator of Equation (12) can be organized so that the exponents are arranged in ascending order, i.e., $\alpha_{k+1} > \alpha_k$ and $\beta_{l+1} > \beta_l$. By using the Euler formula

$$\exp\{jx\} = \cos x + j \sin x \tag{13}$$

the term $j^\alpha$ can be written as

$$j^\alpha = \exp\{j\alpha\pi/2\} = \cos\left(\frac{\alpha\pi}{2}\right) + j \sin\left(\frac{\alpha\pi}{2}\right) \tag{14}$$

Hence, Equation (12) can be reformulated as

$$\varepsilon_r(\omega) = \frac{\mathcal{P}_R(\omega) + j\mathcal{P}_I(\omega)}{\mathcal{Q}_R(\omega) + j\mathcal{Q}_I(\omega)} = \frac{\mathcal{P}_R\mathcal{Q}_R + \mathcal{P}_I\mathcal{Q}_I}{\mathcal{Q}_R^2 + \mathcal{Q}_I^2} - j\frac{\mathcal{P}_R\mathcal{Q}_I - \mathcal{P}_I\mathcal{Q}_R}{\mathcal{Q}_R^2 + \mathcal{Q}_I^2} \tag{15}$$

where

$$\mathcal{P}_R(\omega) = a_0 + \sum_{k=1}^{N} a_k \omega^{\alpha_k} \cos\left(\frac{\alpha_k \pi}{2}\right) \tag{16}$$

$$\mathcal{P}_I(\omega) = \sum_{k=1}^{N} a_k \omega^{\alpha_k} \sin\left(\frac{\alpha_k \pi}{2}\right) \tag{17}$$

and

$$\mathcal{Q}_R(\omega) = b_0 + \sum_{l=1}^{M} b_l \omega^{\beta_l} \cos\left(\frac{\beta_l \pi}{2}\right) \tag{18}$$

$$\mathcal{Q}_I(\omega) = \sum_{l=1}^{M} b_l \omega^{\beta_l} \sin\left(\frac{\beta_l \pi}{2}\right) \tag{19}$$

In order to prevent the Equation (12) from predicting non-physical results, the parameters have to satisfy specific constrains. First of all, it has to reproduce the real behavior as the frequency $\omega$ approaches zero and infinity. By an inspection of Equation (15), it can be inferred that the imaginary part vanishes as $\omega \to 0$ and

$$\varepsilon_{rs} = \frac{a_0}{b_0} \geq 1 \tag{20}$$

Similarly, as $\omega \to \infty$, the imaginary part is canceled only if $\alpha_N = \beta_M$ and consequently,

$$\varepsilon_{r\infty} = \frac{a_N}{b_M} \geq 1 \tag{21}$$

Moreover, the constrain $\varepsilon_{rs} \geq \varepsilon_{r\infty}$ requires the following inequality

$$\frac{a_0}{b_0} \geq \frac{a_N}{b_M} \tag{22}$$

Additionally to reduce numerical instabilities, all the coefficients should satisfy the inequalities

$$a_k, b_l \geq 0 \quad \forall k, l \tag{23}$$

To ensures the passivity, the parameters should be selected in such a way as to satisfy the relationship

$$\text{Im}\{\varepsilon_r(\omega)\} = \frac{\mathcal{P}_R(\omega)\mathcal{Q}_I(\omega) - \mathcal{P}_I(\omega)\mathcal{Q}_R(\omega)}{\mathcal{Q}_R^2(\omega) + \mathcal{Q}_I^2(\omega)} \geq 0 \tag{24}$$

that is $\mathcal{P}_R(\omega)\mathcal{Q}_I(\omega) \geq \mathcal{P}_I(\omega)\mathcal{Q}_R(\omega)$. On the other hand, the real part is always positive if $0 \leq \alpha_k, \beta_l \leq 1$, $\forall k, l$. However, for passive media, the causality implies that $\varepsilon_r(\omega)$ has to be holomorphic in the upper half-plane [80], establishing a relation between real and imaginary parts.

The proposed model offers more flexibility than those based on (8)–(11), and enables a more effective parameterization of arbitrary dispersive media properties, as well as a better fitting capability of the experimental data over broad frequency ranges. It includes the empirical four-parameter dielectric model based on viscoelastic analysis techniques [67], the fractional model illustrated in [81] as well as the mixing Debye and Cole–Cole relationship proposed in [79]. It can be also adapted to include the complex conjugate residual pairs and the Drude critical-points dispersive relationships [82,83], the two-parameter variant of the Drude model presented in [84], as well as the modified Lorentz model [85]. Therefore, the use of (12) makes the modeling of permittivity functions composed of multiple Debye or Cole–Cole terms possible. On the other hand, the power-law nature of the time-harmonic permittivity function allows its incorporation into FDTD schemes, thus enabling the modeling of ultrawideband electromagnetic pulse propagation in arbitrary dispersive media

and biological tissues [48,73,86]. By applying a dedicated optimization algorithm based on the EWQPSO [87,88], the free parameters $a_k, b_l, \alpha_k, \beta_l, N, M$ can be evaluated in such a way as to find the optimal curve that fits the experimental data corresponding to the real and imaginary components of the permittivity spectrum. Thanks to its capability of dealing with any set of data, the proposed method enables great versatility and can reproduce fine details in a very effective way. Compared to other stochastic optimization methods available in the scientific literature, the aforementioned technique exhibits superior effectiveness in terms of precision, reliability, and reduced computational burden.

## 4. FDTD Modeling

The FDTD method is one of the leading numerical tools for transient analysis of complex EM problems. The algorithm is based on the direct time integration of Maxwell's equations in differential form and, from a computational point of view, it is numerically robust, it has well-known sources of numerical errors, and it can be easily implemented. Therefore, FDTD implementation would be advantageous due to its capability to deal with a broadband dielectric response, its ability to produce time-domain movies, its near-linear scalability when run on many processors, and its ability to manage the most varied geometrical and dispersive properties of the material being simulated [89]. However, one of the strengths of FDTD is that it allows the modeling of broadband electromagnetic structures through a single simulation run.

The evolution of the FDTD method has been characterized by the development of various algorithms extending its implementation to different applications. Of great interest is the analysis of transient wave propagation inside dispersive media. In this case, the main challenge associated with the FDTD algorithm regards the numerical treatment of the constitutive relation identified by Equation (2). The available techniques for handling dispersive dielectric media through FDTD-based simulations can be grouped in three main categories: RC methods [90,91], ADE techniques [85,92,93], and ZT methods [94,95]. The RC formulation is an integral approach and its implementation requires the approximation of the convolution integral, corresponding to the time domain constitutive relation, by a discrete summation in combination with a recursive procedure. The ADE technique utilizes ordinary differential equations derived by inverse Fourier transform of the frequency-dependent constitutive relations. These equations are time-stepped in combination with Maxwell's curl equations, yielding a composite self-consistent system. Finally, in the ZT method, the complex permittivity is converted in the z-domain and implemented directly into the FDTD paradigm using existing digital filtering theory. As compared to other methods, the RC approaches are typically faster and require fewer computer memory. However, ADE methods can provide simpler arithmetic implementation and they can be straightforwardly combined with nonlinear dispersive media. Moreover, they have the same second-order accuracy as the piecewise linear recursive convolution method.

For Debye, Drude, Lorentz, or more general rational function-based relative complex permittivity models involving integer function of $j\omega$, the time-domain expression of the constitutive relation can be easily formulated. On the other hand, power law dielectric response involving non-integer power of $j\omega$ leads to fractional derivatives in time domain. Therefore, the implementation of FDTD algorithms for modeling electromagnetic wave propagation is much more complicated since the resulting fractional-time differential equations pose discretization difficulties and are more expensive to solve. With reference to the dielectric response models identified by Equations (8)–(10), some researchers proposed approximations based on Debye functions expansion [96,97]. In this way, the power law-based complex permittivity functions can be efficiently and directly incorporated into the FDTD method. The resulting scheme has the same stability and accuracy properties as the ones used to implement the Debye model, but the polarization dynamics are not accurately represented in the time domain with a fixed accuracy. However, these approaches share the possibility of producing negative weights and/or relaxation frequencies, which have no physical significance. Further, the singularity arising from fractional relaxation is not

taken into account [98]. Other attempts utilize FDTD approach and ZT method for the analysis of wave propagation in CC, CD, and HN dispersive media [95,99,100]. For instance, in [100], the discretization of the fractional derivatives in the time domain was performed by using FILT, to transform the time-harmonic relative permittivity function into a time-domain response, and Prony's method, to extract the parameters in the z-domain. However, these FDTD algorithms may require storing of some auxiliary field quantities leading to large memory requirement. Alternative FDTD implementations based on the convolution integral formulation are presented in [101]. This approach required storing of the electric field and some related auxiliary quantities relevant to the previous time-step only.

Other techniques based on fractional derivatives were proposed in [72,86,98,102–104]. In [72], a time-marching scheme was proposed so to incorporate the general HN response directly in the FDTD algorithm. In accordance with the Riemann–Liouville derivative, the developed formulation was based on the optimal truncation of the binomial series relevant to the HN fractional derivative operator. The resulting second-order finite-difference scheme was applied to the analysis of pulse-wave propagation in dispersive media modeled by CC, CD, and HN dispersive media. It was demonstrated that for $\alpha < 0.5$, $\beta < 0.25$, and $\omega\tau \leq 10$ the developed FDTD scheme allows a reliable and very accurate space-time evaluation of broadband wave propagation in complex layered systems. On the other hand, a reduced numerical accuracy was highlighted for $\alpha > 0.5$, $\beta > 0.25$, and $\omega\tau > 10$. In [98], the authors derived a second-order FDTD algorithm to construct time-domain simulations for HN dielectric response. Applying generalized Gaussian quadratures, the algorithm approximates the polarization convolution induced by the HN model using a sum of exponentials. Moreover, an efficient numerical technique was employed with the aim of performing an update of the equations by considering the information from the previous time step only. In this way, the required storage is reduced from $O(N)$ to $O(\log N)$. Using von Neumann analysis, the authors also demonstrated numerical stability with no additional restriction on the time step. However, numerical validations of the scheme were carried out in 1D free space. An implicit Crank–Nicolson scheme and an explicit leap-frog scheme including the Letnikov fractional derivative were proposed in [102] for solving Maxwell's equations in CC dispersive media. Numerical stability and optimal error estimation were assessed for both schemes. Numerical analyses were carried out, also, in 2D free space. A fully discrete implicit space–time FDTD scheme for Maxwell's equations in a CC dispersive medium was proposed and analyzed in [104]. The proposed scheme combines an implicit Crank–Nicolson method in time domain, and second-order difference operators to approximate spatial differential terms. The relevant Caputo fractional derivative term is approximated using the $L2 - 1_\sigma$ formula and a weighted approach. Therefore, using energy and inductive reasoning methods, the authors proved the unconditionally energy stability of the proposed scheme as well as its convergence with second-order accuracy in both time and space. In this case also, numerical validation was performed both in 2D and 3D free-space domains.

### 4.1. FDTD Scheme

The FDTD algorithms previously recalled suffer from some weaknesses due to their applicability to specific power-law dielectric response. In fact, their effectiveness was tested for studying the free-space electromagnetic field propagation in dispersive media modeled with CC, CD, and HN complex permittivity functions. By virtue of these limitations and with the aim to provide a better parametrization of the dispersive media properties as well as a better broadband fitting of the experimental dielectric response, the authors have developed an FDTD scheme involving the complex permittivity function expressed by Equation (12) [73,86].

The complex relative permittivity function described by (12) can always be rearranged as

$$\varepsilon_r(\omega) = \varepsilon_{rs} \frac{1 + \sum_{k=1}^{N} a_k (j\omega)^{\alpha_k}}{1 + \sum_{l=1}^{M} b_l (j\omega)^{\beta_l}} \tag{25}$$

where the coefficients $a_k, b_l$ are different from those used in (12). For a nonmagnetic dispersive dielectric material with complex relative permittivity described by (25), the time-harmonic Ampere law can be written as:

$$\nabla \times \mathbf{H} = \sigma \mathbf{E} + \mathbf{J} + \mathbf{J}_0 \tag{26}$$

where $\sigma$ is the electric conductivity, $\mathbf{J}_0$ is the source current density, and the auxiliary displacement current density $\mathbf{J}$ satisfies the equation

$$\mathbf{J} = j\omega \varepsilon_0 \varepsilon_{rs} \frac{1 + \sum_{k=1}^{N} a_k (j\omega)^{\alpha_k}}{1 + \sum_{l=1}^{M} b_l (j\omega)^{\beta_l}} \mathbf{E} \tag{27}$$

or

$$\mathbf{J} + \sum_{l=1}^{M} b_l (j\omega)^{v_l} (j\omega)^{\beta_l - v_l} \mathbf{J} = j\omega \varepsilon_0 \varepsilon_{rs} \mathbf{E} + \varepsilon_0 \varepsilon_{rs} \sum_{k=1}^{N} a_k (j\omega)^{v_k} (j\omega)^{1 + \alpha_k - v_k} \mathbf{E} \tag{28}$$

where $v_l$ and $v_k$ are the integers obtained by rounding up the real numbers $\beta_l$ and $1 + \alpha_k$ to their next highest integer, respectively. Taking the inverse Fourier transform of (28) gives

$$\boldsymbol{\mathcal{J}} + \sum_{l=1}^{M} b_l \frac{\partial^{v_l}}{\partial t^{v_l}} \int_0^t \zeta_l(t-u) \boldsymbol{\mathcal{J}}(u) \mathrm{d}u - \varepsilon_0 \varepsilon_{rs} \sum_{k=1}^{N} a_k \frac{\partial^{v_k}}{\partial t^{v_k}} \int_0^t \zeta_k(t-u) \boldsymbol{\mathcal{E}}(u) \mathrm{d}u = \varepsilon_0 \varepsilon_{rs} \frac{\partial \boldsymbol{\mathcal{E}}}{\partial t} \tag{29}$$

where

$$\zeta_l(t) = b_l \int_{-\infty}^{\infty} (j\omega)^{\beta_l - v_l} e^{j\omega t} \mathrm{d}\omega \tag{30}$$

$$\zeta_k(t) = a_k \int_{-\infty}^{\infty} (j\omega)^{1 + \alpha_k - v_k} e^{j\omega t} \mathrm{d}\omega \tag{31}$$

For a real number $\beta > -1$, the gamma function is defined as

$$\Gamma(\beta + 1) = \int_0^{\infty} t^{\beta} e^{-t} \mathrm{d}t \tag{32}$$

and the Fourier transform of the power law function is

$$\mathcal{F}\{t^{\beta}\} = \int_0^{\infty} t^{\beta} e^{-j\omega t} \mathrm{d}t \tag{33}$$

where $t^{\beta} = 0$ for $t < 0$. Thus, using the variable change $u = j\omega t$, it is straightforward to obtain

$$\mathcal{F}\{t^{\beta}\} = (j\omega)^{-\beta - 1} \int_0^{\infty} u^{\beta} e^{-u} \mathrm{d}u = (j\omega)^{-\beta - 1} \Gamma(\beta + 1) \tag{34}$$

The inverse Fourier transform of (34) gives

$$\mathcal{F}^{-1}\{\mathcal{F}\{t^{\beta}\}\} = t^{\beta} = \Gamma(\beta + 1) \int_0^{\infty} (j\omega)^{-\beta - 1} e^{j\omega t} \mathrm{d}\omega \tag{35}$$

and comparing (35) with (30)–(31) it could be achieved

$$\zeta_l(t) = b_l \frac{t^{v_l - \beta_l - 1}}{\Gamma(v_l - \beta_l)} \tag{36}$$

$$\zeta_k(t) = a_k \frac{t^{v_k - \alpha_k - 2}}{\Gamma(v_k - \alpha_k - 1)} \tag{37}$$

As a result, the Ampere's law in time domain can be written as

$$\nabla \times \boldsymbol{\mathcal{H}} = \sigma \boldsymbol{\mathcal{E}} + \boldsymbol{\mathcal{J}} + \boldsymbol{\mathcal{J}}_0 \tag{38}$$

where the auxiliary displacement current density $\mathcal{J}$ satisfies the equation

$$\mathcal{J} + \sum_{l=1}^{M} b_l \mathscr{D}_t^{\beta_l} \mathcal{J} - \varepsilon_0 \varepsilon_{rs} \sum_{k=1}^{N} a_k \mathscr{D}_t^{1+\alpha_k} \mathcal{E} = \varepsilon_0 \varepsilon_{rs} \frac{\partial \mathcal{E}}{\partial t} \tag{39}$$

In Equation (39), $\mathscr{D}_t^{\beta_l}$ and $\mathscr{D}_t^{1+\alpha_k}$ are the fractional derivative operators of order $\beta_l$ and $1 + \alpha_k$, respectively, given by

$$\mathscr{D}_t^{\beta_l} \mathcal{J} = \frac{\partial^{v_l}}{\partial t^{v_l}} \int_0^t \frac{(t-u)^{v_l - \beta_l - 1}}{\Gamma(v_l - \beta_l)} \mathcal{J}(u) \mathrm{d}u \tag{40}$$

$$\mathscr{D}_t^{1+\alpha_k} \mathcal{E} = \frac{\partial^{v_k}}{\partial t^{v_k}} \int_0^t \frac{(t-u)^{v_k - \alpha_k - 2}}{\Gamma(v_k - \alpha_k - 1)} \mathcal{E}(u) \mathrm{d}u \tag{41}$$

Applying a second-order finite-difference scheme at the time instant $t = m\Delta t$, from (38) it follows that

$$\boldsymbol{\nabla} \times \mathcal{H}|^m = \sigma \mathcal{E}|^m + \mathcal{J}|^m + \mathcal{J}_0|^m \tag{42}$$

The right hand of Equation (42) can be evaluated using the semi-implicit approximation

$$\mathcal{J}|^m = \frac{\mathcal{J}|^{m-\frac{1}{2}} + \mathcal{J}|^{m+\frac{1}{2}}}{2} \tag{43}$$

$$\mathcal{E}|^m = \frac{\mathcal{E}|^{m-\frac{1}{2}} + \mathcal{E}|^{m+\frac{1}{2}}}{2} \tag{44}$$

as well as the numerical approximation

$$\frac{\partial \mathcal{E}}{\partial t}\bigg|^m = \frac{\mathcal{E}|^{m+\frac{1}{2}} - \mathcal{E}|^{m-\frac{1}{2}}}{\Delta t} \tag{45}$$

with $\Delta t$ as the time step. In particular, evaluating (39) at the time step $t = m\Delta t$ yields

$$\mathcal{E}|^{m+\frac{1}{2}} = \mathcal{E}|^{m-\frac{1}{2}} + \frac{\Delta t}{\varepsilon_0 \varepsilon_{rs}} \left[ \frac{\mathcal{J}|^{m-\frac{1}{2}} + \mathcal{J}|^{m+\frac{1}{2}}}{2} + \sum_{l=1}^{M} b_l \mathscr{D}_t^{\beta_l} \mathcal{J}\bigg|^m \right] - \Delta t \sum_{k=1}^{N} a_k \mathscr{D}_t^{1+\alpha_k} \mathcal{E}\bigg|^m \tag{46}$$

An inspection of (46) makes evident the need to calculate the finite-difference approximation of the fractional derivative of both $\mathcal{J}$ and $\mathcal{E}$. To this end, by setting

$$\mathcal{I}_l = \int_0^t (t-u)^{v_l - \beta_l - 1} \mathcal{J}(u) \mathrm{d}u \tag{47}$$

and applying the central finite difference approximation at the time instant $t = m\Delta t$, it can be obtained

$$\mathcal{I}_l|^m \approx \sum_{p=0}^{m-1} \mathcal{J}|^{m-p-\frac{1}{2}} \int_{p\Delta t}^{(p+1)\Delta t} u^{v_l - \beta_l - 1} \mathrm{d}u = \frac{\Delta t^{v_l - \beta_l}}{v_l - \beta_l} \sum_{p=0}^{m-1} \left[ (p+1)^{v_l - \beta_l} - p^{v_l - \beta_l} \right] \mathcal{J}|^{m-p-\frac{1}{2}} \tag{48}$$

By using the following exponential expansion of order $Q_l$ [105]

$$(p+1)^{v_l - \beta_l} - p^{v_l - \beta_l} \approx \sum_{q=1}^{Q_l} r_{l,q} e^{-s_{l,q} p} \tag{49}$$

with $r_{l,q}, s_{l,q}$ as suitable coefficient minimizing the mean square error function, and upon setting

$$\Psi_{l,q}\bigg|^m = \sum_{p=0}^{m-1} r_{l,q} e^{-s_{l,q} p} \mathcal{J}|^{m-p-\frac{1}{2}} = r_{l,q} \mathcal{J}|^{m-\frac{1}{2}} + e^{-s_{l,q}} \Psi_{l,q}\bigg|^{m-1} \tag{50}$$

Equation (43) can be rewritten as:

$$\mathcal{I}_l|^m \approx \frac{\Delta t^{v_l - \beta_l}}{v_l - \beta_l} \left( S_l \, \mathcal{J}|^{m-\frac{1}{2}} + \sum_{q=1}^{Q_l} e^{-s_{l,q}} \, \Psi_{l,q}\Big|^{m-1} \right) \tag{51}$$

where

$$S_l = \sum_{q=1}^{Q_l} r_{l,q} \tag{52}$$

Evaluating Equation (40) at $t = m\Delta t$ yields

$$\mathscr{D}_t^{\beta_l} \mathcal{J}\Big|^m = \frac{1}{\Gamma(v_l - \beta_l)} \frac{\partial^{v_l} \mathcal{I}_l}{\partial t^{v_l}}\Big|^m \approx \frac{1}{\Gamma(v_l - \beta_l)\Delta^{v_l}} \left[ \mathcal{I}_l|^{m+1} + \sum_{r=1}^{v_l} (-1)^r \binom{v_l}{r} \mathcal{I}_l|^{m-r+1} \right] \tag{53}$$

and using (46)

$$\mathscr{D}_t^{\beta_l} \mathcal{J}\Big|^m \approx \frac{\Delta t^{-\beta_l}}{(v_l - \beta_l)\Gamma(v_l - \beta_l)} \left\{ S_l \left[ \mathcal{J}|^{m+\frac{1}{2}} + \sum_{r=1}^{v_l} (-1)^r \binom{v_l}{r} \mathcal{J}|^{m-r+\frac{1}{2}} \right] \right.$$
$$\left. + \sum_{r=0}^{v_l} (-1)^r \binom{v_l}{r} \sum_{q=1}^{Q_l} e^{-s_{l,q}} \, \Psi_{l,q}\Big|^{m-r} \right\} \tag{54}$$

or in a more compact form

$$\mathscr{D}_t^{\beta_l} \mathcal{J}\Big|^m \approx \frac{\Delta t^{-\beta_l}}{\Gamma(1 + v_l - \beta_l)} \sum_{r=0}^{v_l} \left[ (-1)^r \binom{v_l}{r} S_l \, \mathcal{J}|^{m-r+\frac{1}{2}} + \sum_{q=1}^{Q_l} (-1)^r \binom{v_l}{r} e^{-s_{l,q}} \, \Psi_{l,q}\Big|^{m-r} \right] \tag{55}$$

The same procedure can be applied to Equation (41). In detail, by setting

$$\mathcal{I}_k = \int_0^t (t-u)^{v_k - \alpha_k - 2} \mathcal{E}(u) \mathrm{d}u \tag{56}$$

using the exponential expansion

$$(p+1)^{v_k - \alpha_k - 1} - p^{v_k - \alpha_k - 1} \approx \sum_{q=1}^{Q_k} \tilde{r}_{k,q} e^{-\tilde{s}_{k,q} p} \tag{57}$$

as well as

$$\Phi_{k,q}\Big|^m = \tilde{r}_{k,q} \, \mathcal{E}|^{m-\frac{1}{2}} + e^{-\tilde{s}_{k,q}} \, \Phi_{k,q}\Big|^{m-1} \tag{58}$$

it can be readily obtained at the time instant $t = m\Delta t$

$$\mathscr{D}_t^{1+\alpha_k} \mathcal{E}\Big|^m \approx \frac{\Delta t^{-(\alpha_k + 1)}}{(v_k - \alpha_k - 1)\Gamma(v_k - \beta_k - 1)} \left\{ S_k \left[ \mathcal{E}|^{m+\frac{1}{2}} + \sum_{r=1}^{v_k} (-1)^r \binom{v_k}{r} \mathcal{E}|^{m-r+\frac{1}{2}} \right] \right.$$
$$\left. + \sum_{r=0}^{v_k} (-1)^r \binom{v_k}{r} \sum_{q=1}^{Q_k} e^{-\tilde{s}_{k,q}} \, \Phi_{k,q}\Big|^{m-r} \right\} \tag{59}$$

or

$$\mathscr{D}_t^{1+\alpha_k} \mathcal{E}\Big|^m \qquad \approx \qquad \frac{\Delta t^{-(\alpha_k + 1)}}{\Gamma(v_k - \beta_k)} \sum_{r=0}^{v_k} \left[ (-1)^r \binom{v_k}{r} S_k \mathcal{E}|^{m-r+\frac{1}{2}} + \sum_{q=1}^{Q_k} (-1)^r \binom{v_k}{r} e^{-\tilde{s}_{k,q}} \, \Phi_{k,q}\Big|^{m-r} \right] \tag{60}$$

with

$$S_k = \sum_{q=1}^{Q_k} \tilde{r}_{k,q} \tag{61}$$

On the basis of (59), it can be readily obtained that

$$\sum_{k=1}^{N} a_k \mathscr{D}_t^{1+\alpha_k} \boldsymbol{\mathcal{E}}\Big|^m \approx \frac{1}{\Delta t} \sum_{k=1}^{N} B_k S_k \boldsymbol{\mathcal{E}}\Big|^{m+\frac{1}{2}} + \frac{1}{\Delta t} \sum_{k=1}^{N} B_k S_k \sum_{r=1}^{v_k} (-1)^r \binom{v_k}{r} \boldsymbol{\mathcal{E}}\Big|^{m-r+\frac{1}{2}}$$

$$+ \frac{1}{\Delta t} \sum_{k=1}^{N} B_k \sum_{r=0}^{v_k} (-1)^r \binom{v_k}{r} \sum_{q=1}^{Q_k} e^{-\tilde{s}_{k,q}} \Phi_{k,q}\Big|^{m-r} \tag{62}$$

where

$$B_k = \frac{a_k \Delta t^{-\alpha_k}}{\Gamma(v_k - \beta_k)} \tag{63}$$

Combining (62) with (46),

$$A\boldsymbol{\mathcal{E}}\Big|^{m+\frac{1}{2}} = \boldsymbol{\mathcal{E}}\Big|^{m-\frac{1}{2}} + \frac{\Delta t}{\varepsilon_0 \varepsilon_{rs}} \left[ \frac{\boldsymbol{\mathcal{J}}\Big|^{m-\frac{1}{2}} + \boldsymbol{\mathcal{J}}\Big|^{m+\frac{1}{2}}}{2} + \sum_{l=1}^{M} b_l \mathscr{D}_t^{\beta_l} \boldsymbol{\mathcal{J}}\Big|^m \right]$$

$$- \sum_{k=1}^{N} B_k S_k \sum_{r=1}^{v_k} (-1)^r \binom{v_k}{r} \boldsymbol{\mathcal{E}}\Big|^{m-r+\frac{1}{2}} - \sum_{k=1}^{N} B_k \sum_{r=0}^{v_k} (-1)^r \binom{v_k}{r} \sum_{q=1}^{Q_k} e^{-\tilde{s}_{k,q}} \Phi_{k,q}\Big|^{m-r} \tag{64}$$

with

$$A = 1 + \sum_{k=1}^{N} B_k S_k \tag{65}$$

Moreover, combining (64) with (44),

$$\boldsymbol{\mathcal{E}}\Big|^m = \frac{1+A}{2A} \boldsymbol{\mathcal{E}}\Big|^{m-\frac{1}{2}} + \frac{\Delta t}{2A\varepsilon_0 \varepsilon_{rs}} \left[ \frac{\boldsymbol{\mathcal{J}}\Big|^{m-\frac{1}{2}} + \boldsymbol{\mathcal{J}}\Big|^{m+\frac{1}{2}}}{2} + \sum_{l=1}^{M} b_l \mathscr{D}_t^{\beta_l} \boldsymbol{\mathcal{J}}\Big|^m \right]$$

$$- \frac{1}{2A} \sum_{k=1}^{N} B_k S_k \sum_{r=1}^{v_k} (-1)^r \binom{v_k}{r} \boldsymbol{\mathcal{E}}\Big|^{m-r+\frac{1}{2}} - \frac{1}{2A} \sum_{k=1}^{N} B_k \sum_{r=0}^{v_k} (-1)^r \binom{v_k}{r} \sum_{q=1}^{Q_k} e^{-\tilde{s}_{k,q}} \Phi_{k,q}\Big|^{m-r} \tag{66}$$

Therefore, by combining (26) with (43) and (66), one can readily obtain the equation providing $\boldsymbol{\mathcal{J}}\big|^{m+\frac{1}{2}}$ as a function of the fields calculated in the previous steps. Once this current density term is evaluated, the electric field distribution can be derived from (42) as

$$\boldsymbol{\mathcal{E}}\Big|^{m+\frac{1}{2}} = -\boldsymbol{\mathcal{E}}\Big|^{m-\frac{1}{2}} + \frac{2}{\sigma} \boldsymbol{\nabla} \times \boldsymbol{\mathcal{H}}\Big|^m - \frac{\boldsymbol{\mathcal{J}}\Big|^{m-\frac{1}{2}} + \boldsymbol{\mathcal{J}}\Big|^{m+\frac{1}{2}}}{2} - \boldsymbol{\mathcal{J}}_0\Big|^m \tag{67}$$

Finally, the following update equation for the magnetic field can be obtained

$$\boldsymbol{\mathcal{H}}\Big|^{m+1} = \boldsymbol{\mathcal{H}}\Big|^m - \frac{\Delta t}{\mu_0} \boldsymbol{\nabla} \times \boldsymbol{\mathcal{E}}\Big|^{m+\frac{1}{2}} \tag{68}$$

Dedicated UPML boundary conditions were derived and numerically implemented accounting for the electrical conductivity and the general fractional polynomial series approximation expressed by Equation (12). The mathematical procedure is well detailed in [72,73]. In particular, the stretched auxiliary electric field and density current vectors involving the fractional derivative equation were suitably combined. The resulting equations on the Yee lattice were derived by adopting the usual leapfrog scheme in time and the semi-implicit approximation. In particular, in accordance with the complex coordi-

nate stretching approach, the update equations for both electric and magnetic field within the UPML termination were derived. Moreover, the UPML parameters was heuristically optimized with the aim to enhance the absorption of electromagnetic waves, thus minimizing the spurious reflection level in the solution domain. However, another main issue characterizing our formulation regards the nonlinear fitting, expressed by Equation (49), needed in the finite-difference approximation of the fractional derivative operator. This topic was addressed by using the EWQPSO, too. The overall algorithm is shown to be second-order accuracy, and it satisfies the standard FDTD stability condition. In particular, following von Neumann's spectral approach, it was investigated whether the CFL stability condition was satisfied [89,106]. Due to the power-law dielectric function and the corresponding approximation of fractional derivatives, this study required special treatment. In [73], we derived the characteristic polynomial equation for the fractional-calculus-based FDTD algorithm demonstrating that the corresponding roots lie inside the unit circle in the z-plane. In particular, assuming the the 1D case and time-harmonic dependence of the electromagnetic field, the field vectors $\mathcal{E}$, $\mathcal{H}$, and $\mathcal{J}$ can be Fourier-transformed with respect to the space variable so that the first-order derivatives in time and space degenerate in algebraic multiplication operators. After some calculation, well detailed in [73], the following characteristic polynomial equation can be derived:

$$c_0^2(\Delta t)^2\Theta^2(\xi)\tilde{\mathcal{D}}_r(z)z^2 + (z^2-1)^2\tilde{\mathcal{N}}_r(z) = 0 \tag{69}$$

where

$$\Theta(\xi) = \xi\mathrm{sinc}\frac{\xi\Delta x}{2} \tag{70}$$

$$z = \exp\left(\frac{j\omega\Delta t}{2}\right) \tag{71}$$

where $c_0$ is the speed of light in free space, $\Delta x$ and $\xi$—the spatial increment of the grid and the real wavenumber of the arbitrary harmonic field component, respectively. Moreover, $\tilde{\mathcal{D}}_r$ and $\tilde{\mathcal{N}}_r$ are, respectively, the numerator and denominator polynomials of the transformed permittivity function as a rational function of $z$. The calculated eigenvalues satisfied the following condition

$$\max_i |z_i| \leq 0 \tag{72}$$

for any spatial frequency $\xi \in [0, \pi/\Delta x]$ once the Courant factor $S = c_0\Delta t/\Delta x \leq 1$ is selected, thus verifying the numerical stability of the proposed time-marching scheme.

### 4.2. Applications

The reliability of the developed FDTD procedure was assessed with reference to the ultrawideband electromagnetic pulse propagation in single and multilayered 1D systems involving dielectric materials exhibiting a power law response. In [72], we studied the wave propagation focusing the attention on UPML characteristics and checking that the numerical results do not feature any long time divergence. On the other hand, in [73], we investigated the numerical stability of the FDTD algorithm, as well as the optimization procedure adopted to perform the nonlinear fitting of the exponential terms appearing in the finite-difference approximation of the fractional derivative operator.

A relevant stress test aimed to demonstrate the effectiveness of the proposed algorithm regards the general layered structure, illustrated in Figure 1, involving dielectric materials with complex permittivity modeled by Equation (12). All the investigations were performed implementing the TFSF formulation, and a sinusoidally time-modulated Gaussian pulse source covering a spectral bandwidth from 100 MHz to 10 GHz. Figure 1 also highlights the plane wave excitation with electric field linearly polarized along the $z$–axis and located at a given point $x = x_s$.

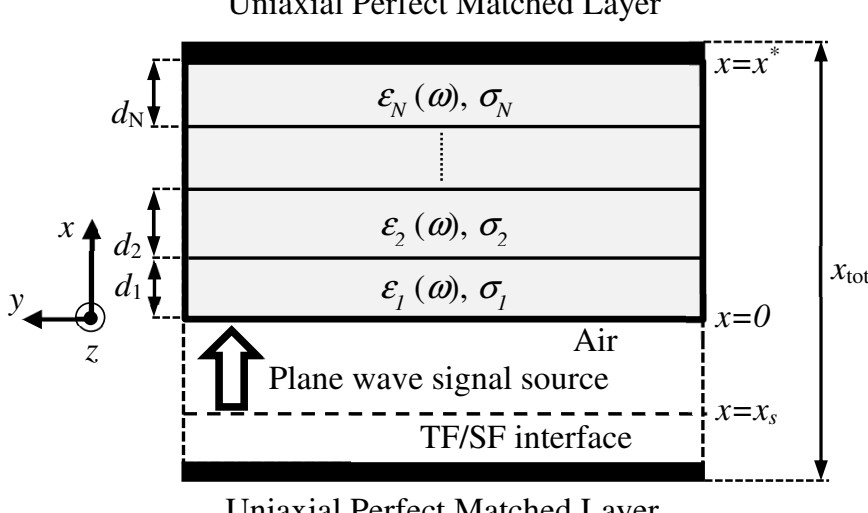

**Figure 1.** Sketch of the layered dielectric system employed in the simulations.

The first test case is relevant to a single-layer structure with thickness $d_1 = 10\,\text{mm}$, electric conductivity $\sigma_1 = 0.035\,\text{S}\,\text{m}^{-1}$, and time-harmonic relative permittivity function

$$\varepsilon_r(\omega) = 2 + 60 \frac{1 + (j\omega\tau)^{0.2}}{1 + 9(j\omega\tau)^{0.3} + 2(j\omega\tau)^{0.5} + 10(j\omega\tau)^{0.9}} \tag{73}$$

with relaxation time $\tau = 318\,\text{ps}$. The exponents and coefficients characterizing the polynomials involved in (73) were arbitrarily chosen. Starting from the space-time field distribution, the reflectance and transmittance spectra were evaluated and compared with that calculated using the analytical technique based on the transfer matrix approach (see Figure 2) [107].

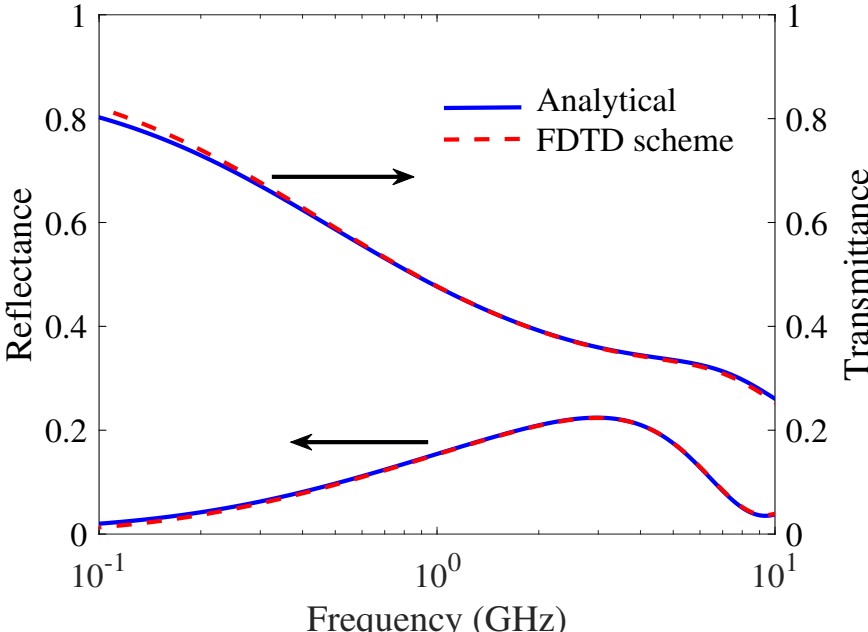

**Figure 2.** Reflectance and transmittance of the dielectric slab versus the frequency.

The second test case is more general since it involves a three-layer dispersive dielectric slab. The geometric and electrical parameters are reported in Table 1 and the complex relative permittivities are

$$\varepsilon_{r1}(\omega) = 2.4 + 26\frac{1 + 2(j\omega\tau_1)^{0.2}}{1 + 5(j\omega\tau_1)^{0.03} + 6.54(j\omega\tau_1)^{0.14} + 0.2(j\omega\tau_1)^{0.8} + 13(j\omega\tau_1)^{0.9}} \tag{74}$$

$$\varepsilon_{r2}(\omega) = 4 + 44\frac{1 + 3(j\omega\tau_2)^{0.3} + 2(j\omega\tau_2)^{0.4}}{1 + 5(j\omega\tau_2)^{0.14} + 8(j\omega\tau_2)^{0.7} + 6(j\omega\tau_2)^{0.8} + 2(j\omega\tau_2)^{0.83}} \tag{75}$$

$$\varepsilon_{r3}(\omega) = 6 + 60\frac{1 + 5(j\omega\tau_3)^{0.3}}{1 + 4(j\omega\tau_3)^{0.29} + 5(j\omega\tau_3)^{0.6} + 6(j\omega\tau_2)^{0.8}} \tag{76}$$

**Table 1.** Geometric and electrical parameters of the three-layer dielectric slab.

| Layer | Thickness (mm) | Conductivity (S m$^{-1}$) | Relaxation Time (ps) |
|---|---|---|---|
| 1 | 20 | $3.5 \times 10^{-2}$ | 16 |
| 2 | 30 | $5 \times 10^{-2}$ | 5.91 |
| 3 | 20 | $3 \times 10^{-2}$ | 59.1 |

Additionally, in this case, the polynomials in the relationships (74)–(76) were arbitrarily chosen. The reflectance and transmittance spectra were evaluated using the FDTD procedure, and the analytical approach is reported in Figure 3. The excellent agreement with the analytical technique fully validates the developed FDTD methodology. However, the numerical simulations were performed over a time interval more than 300 times the duration of the source pulse, showing a well-behaved distribution and asymptotically decreasing energy over time.

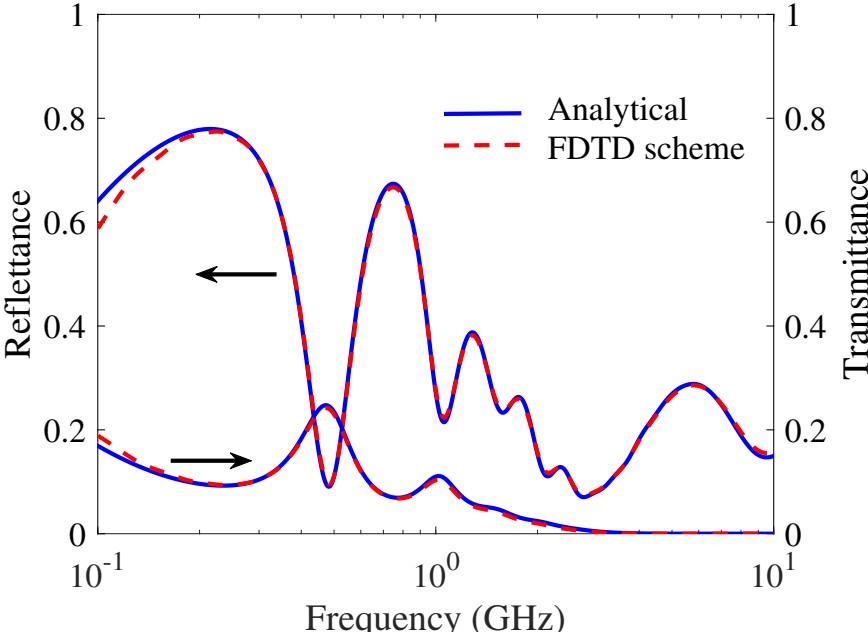

**Figure 3.** Reflectance and transmittance of the three-layer dielectric slab as a function of frequency.

The FDTD scheme was also applied to evaluate the pulse propagation inside dielectrics characterized by a spatially non-homogeneous permittivity exhibiting a power-law frequency dependence [86]. In particular, considering the general slab model illustrated in

Figure 1, a half-space structure filled up with dispersive and non-homogeneous media was considered. The complex relative permittivity function was

$$\varepsilon_r(x,\omega) = \varepsilon_\infty(x) + \frac{\varepsilon_s(x) - \varepsilon_\infty(x)}{1 + (j\omega\tau)^{0.45} + 0.13(j\omega\tau)^{0.75}} \tag{77}$$

where the spatial dependence of the material parameters was expressed through the following Maxwell–Garnet mixing formula [108]

$$\varepsilon_s(x) = 20 \left[ 1 + \frac{f(x)}{3} \frac{\sum\limits_{i=i}^{3} \dfrac{19}{1 + 19N_i}}{1 - \dfrac{f(x)}{3} \sum\limits_{i=1}^{3} \dfrac{19N_i}{1 + 19N_i}} \right] \tag{78}$$

and

$$\varepsilon_\infty(x) = 2 \left[ 1 + \frac{f(x)}{3} \frac{\sum\limits_{i=i}^{3} \dfrac{4}{1 + 4N_i}}{1 - \dfrac{f(x)}{3} \sum\limits_{i=1}^{3} \dfrac{4N_i}{1 + 4N_i}} \right] \tag{79}$$

with the space-dependent filling factor function given by

$$f(x) = f(0) \exp\left\{ -\frac{3x}{d} \right\} \tag{80}$$

In Equations (78)–(80), the initial value of the filling factor, the thickness of the half space and the depolarization factor are $f(0) = 0.3$, $d = 8\,\text{cm}$, $N_1 = 0$, $N_2 = N_3 = 0.5$, respectively. By an inspection of the FDTD numerical results reported in Figure 4, the transmission as well as the reflection phenomenon occurring at the air-dispersive media interface can be noticed. Moreover, the pulse spreading inside the dispersive material and the spatial variation of the group velocity are evident. In [86], we also compared the Maxwell–Garnet and Bruggeman models highlighting differences in the propagation characteristics, especially at low frequency. Such results confirm the need to choose the most appropriate mixing model, since for denser composites, Bruggeman's formula is better suited than the Maxwell–Garnett one.

To prove how meddlesome the developed FDTD technique is, we also studied the electromagnetic scattering problem characterizing the GPR applications [109]. A variety of methods have been proposed to numerically solve the equations governing the GPR problem [110]. However, to increase the effectiveness of GPR numerical modeling, a more general approach is needed, especially considering that the dielectric response of many soils and rocks can be modeled using empirical relationships based on fractional power law. In order to meet this challenge, we extended to the 2D space the general 1D FDTD scheme derived by using fractional derivative theory. Moreover, dedicated 2D UPML boundary conditions were derived and implemented in combination with the basic time-marching scheme. In [40], we applied the resulting numerical algorithm to simulate the scattering from lossy circular cylinder buried in a dispersive soil. Figure 5 shows a sketch of the considered electromagnetic problem highlighting the distance $h_1$ between the electromagnetic source and the vacuum-dispersive soil interface, as well as the depth $h$ within the soil.

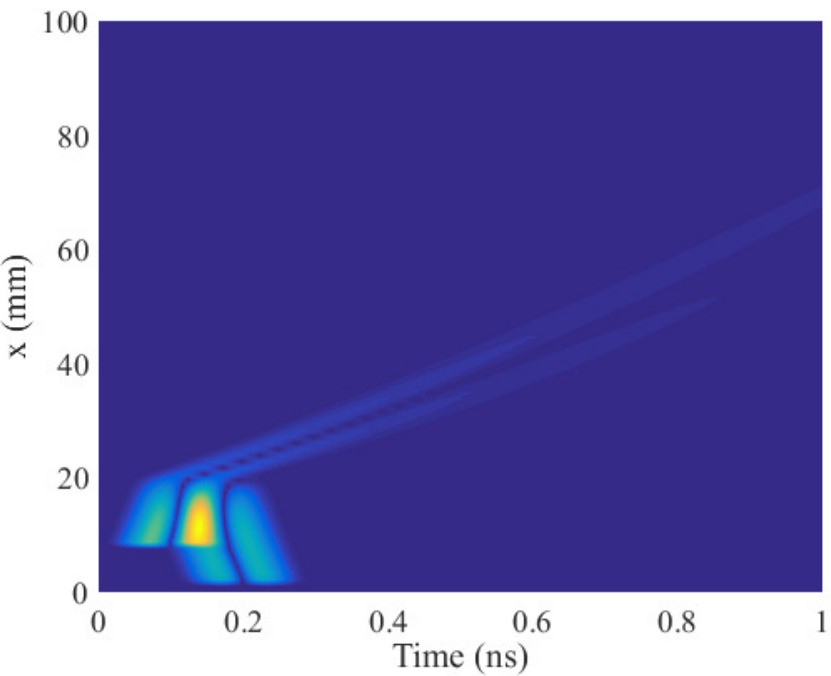

**Figure 4.** Space-time electric field distribution inside a dispersive media modeled by a Maxwell–Garnet relationship characterized by an exponential filling factor function. Reproduced under the terms of a Creative Commons Attribution 4.0 International License [86]. Copyright 2016, The Authors, published by Hindawi Ltd.

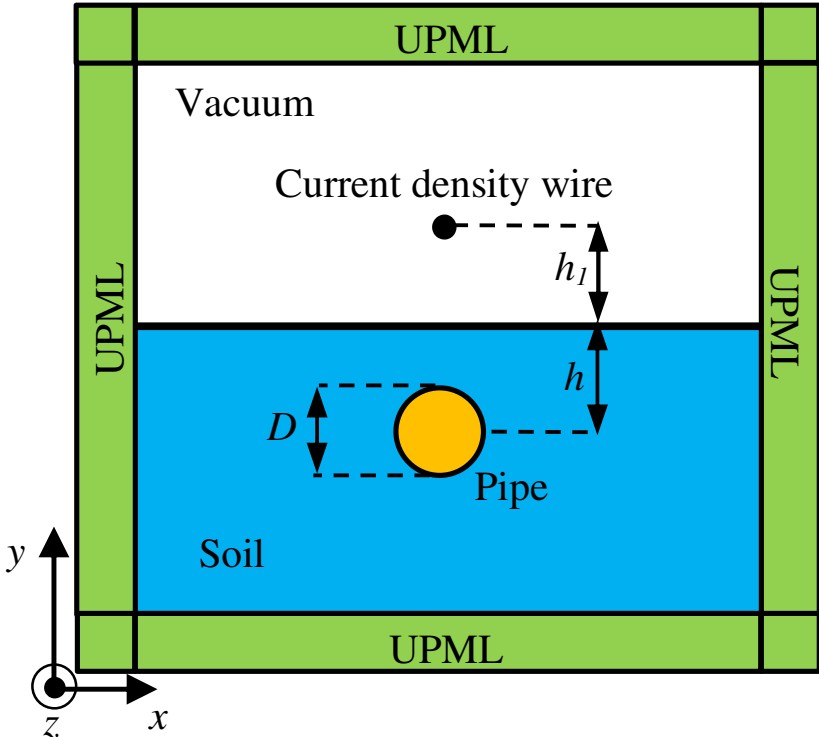

**Figure 5.** Geometrical sketch of the electromagnetic scattering problem.

The frequency-dependent permittivity function of the soil is

$$\varepsilon_r(\omega) = 2 + \frac{4.9}{\varrho_1(\omega)} + \frac{2}{\varrho_2(\omega)} \tag{81}$$

where

$$\varrho_1(\omega) = 1 + 15.1(j\omega\tau_1)^{0.42} + 6.6(j\omega\tau_1)^{0.47} + 11.9(j\omega\tau_1)^{0.58} + 116.7(j\omega\tau_1)^{0.52} \tag{82}$$

$$\varrho_2(\omega) = 1 + 1.43(j\omega\tau_2)^{0.55} \tag{83}$$

and $\tau_1 = 0.13\,\text{ns}$, $\tau_2 = 5.6\,\text{µs}$. The coefficients and power-law exponents characterizing Equations (81)–(83) were recovered by curve fitting of soil experimental data in the frequency range from 10 MHz to 1 GHz [111]. Moreover, the metallic pipe has a conductivity $\sigma_p = 1 \times 10^8\,\text{S m}^{-1}$. Figure 6 shows the calculated B-scan GPR image obtained by collecting the scattered electric field along a straight path on top of the air-dispersive soil surface, when the current density wire and the pipe are placed at $h_1 = 15\,\text{cm}$ and $h = 50\,\text{cm}$, respectively. Inspecting the obtained results, it is clear that the main contribution is due to the direct reflection of the electromagnetic pulse by the pipe. Moreover, in the region bounded by the dashed white curve, the contribution provided by the high-order reflected waves can be inferred.

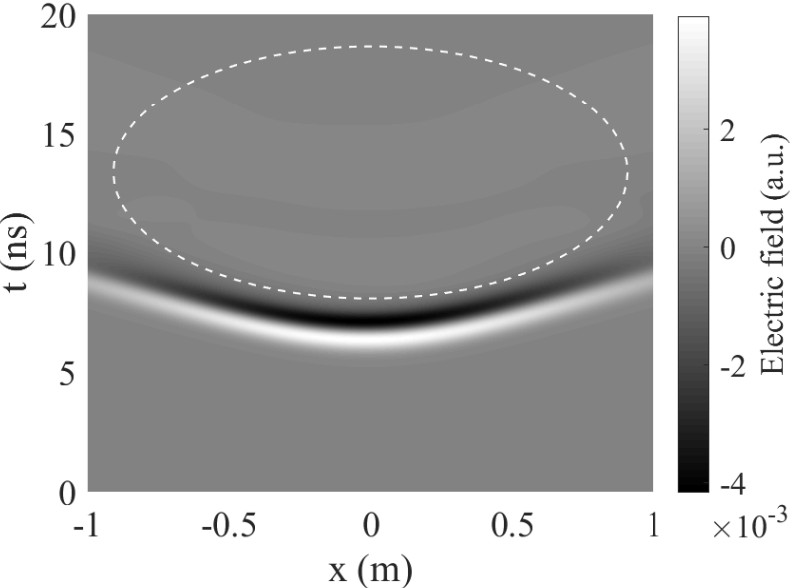

**Figure 6.** B-scan GPR image when the electromagnetic source is placed 15 cm above the air-dispersive soil surface and the pipe is buried 50 cm under the same surface. Reproduced with permission [6]. Copyright 2018, John Wiley and Sons.

## 5. Conclusions

Fractional calculus has been successfully implemented into electromagnetic field theory leading to the generalizations of the Maxwell's equations. The research of numerical techniques for solving FO Maxwell's equations is currently in its early stage. However, it could be an interesting tool for understanding the electromagnetic wave propagation and radiation in dielectrics characterized by power-law response as well as for designing new artificial materials allowing for exotic electromagnetic phenomena. In view of such a fascinating picture, in this paper, we presented a review of FC methods in electromagnetic theory as well as their application to dielectric relaxation modeling. The different topics and numerical techniques regarding the handling of dispersive dielectric media through FDTD-based simulations have been discussed focusing the attention on RC, ADE, and ZT methods. Moreover, the recent developments on the FC-based FDTD algorithm have been illustrated from a mathematical and computational point of view. The key issues concerning the establishment of efficient UPML boundary conditions, stable FDTD scheme, and reduced numerical dispersion have been highlighted. Moreover, the soundness and reliability of the FC-based FDTD scheme were assessed on the basis of some numerical results pertaining to 1D and 2D electromagnetic problems involving dielectric materials exhibiting a power-law response.

**Author Contributions:** Conceptualization, L.M.; methodology, L.M., P.B. and D.C.; data curation, L.M. and P.B.; writing—original draft preparation, L.M.; writing—review and editing, L.M. and D.C.; visualization, L.M. and P.B.; supervision, D.C. All authors have read and agreed to the published version of the manuscript.

**Funding:** This research received no external funding.

**Institutional Review Board Statement:** Not applicable.

**Informed Consent Statement:** Not applicable.

**Conflicts of Interest:** The authors declare no conflict of interest.

## Abbreviations

The following abbreviations are used in this manuscript:

| | |
|---|---|
| ADE | Auxiliary Differential Equation |
| CC | Cole–Cole |
| CD | Cole–Davidson |
| CFL | Courant–Friedrichs–Lewy |
| EM | Electromagnetic |
| EMM | Electromagnetic Metamaterial |
| EWQPSO | Enhanced Weighted Quantum Particle Swarm Optimization |
| FC | Fractional Calculus |
| FDTD | Finite Difference Time Domain |
| FO | Fractional Order |
| FILT | Fast Inverse Laplace Transform |
| FVC | Fractional Vector Calculus |
| GPR | Ground Penetrating Radar |
| HN | Havriliak–Negami |
| PDE | Partial Differential Equation |
| PSO | Particle Swarm Optimization |
| RC | Recursive Convolution |
| TF | Time Fractional |
| TFSF | Total-Field/Scattered-Field |
| TRD | Temporal Rrelaxation Distribution |
| UPML | Uniaxial Perfectly Matched Layer |
| ZT | Z-transform |

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
