# Peer review of "FDTD-Based Electromagnetic Modeling of Dielectric Materials with Fractional Dispersive Response"

_electronics, doi:10.3390/electronics11101588_

Round 1

Reviewer 1 Report

The presented manuscript is, in my opinion, a textbook, not an article. In truth, it is a review, but the description of the FDTD numerical method has been vaguely presented.  The introduction is extensive and the review of the methods used so far has been presented, but in my opinion there are too many formulas and theoretical descriptions in the manuscript. There is very little research presented. No other analysis cases are presented. Finally, the topic ends. Summary is missing. In my opinion, it is important to change the manuscript by adding at least more numerical analysis performed by the FDTD method. 

Author Response

Most likely a misunderstanding occurred since the proposed paper is a review article. In fact, it is not our intention to provide here novel results of our research activities. The aim of this paper is to provide a review of fractional calculus methods in electromagnetic theory as well as their application to dielectric relaxation modeling. However, we agree with the reviewer on the issue regarding the lack of a summary. To this aim, a new section “Conclusions” was added in the revised paper.

Reviewer 2 Report

In the paper 'FDTD-Based Electromagnetic Modeling of Dielectric Materials with Fractional Dispersive Response' the authors present a review of the use of fractional derivatives and integrals. In my opinion, the paper is well organized and timely. The introduction is clear as where as the discussion given by the authors. For all these reasons, I recommend the publication of this manuscript. 

Author Response

We thank the reviewer for his/her positive feedback

Reviewer 3 Report

The authors of the manuscript entitled "FDTD-Based Electromagnetic Modeling of Dielectric Materials with Fractional Dispersive Response" overview the fractional calculus models of dielectric response, propose the fractional polynomial-based dielectric function and thereupon derive fractional Maxwell's equations. As a welcome result, the calculations performed using the proposed FDTD scheme have been verified by accurate analytical calculations. Regretfully, the authors do not provide evidence of superiority of their approach over standard numerical solution of conventional Maxwell's equations. Neither do they support fractional calculations by experimental measurements. Together with the extreme difficulties, cumbersomeness, and computational demands in realizing fractional calculus in simulations, this casts doubts on practical usefulness of the approach proposed.

When reviewing literature on fractional calculus in dielectrics behavior, the authors causelessly ignore works of pioneers in the application of fractional calculus in the dielectric theory, Yu. Kalmykov and R. Hilfer. A number of recent studies of other authors within this area have been disregarded, too.

In the Applications section, it is not clear where Eqs. (73)-(77) have been derived from. Are the polynomials in these relationships arbitrarily chosen or adjusted to fit the spectra?

The language of the manuscript requires certain improvement, too. This especially concerns the Applications section, where several cases of inappropriate wording have been noticed. In addition, there are unexplained abbreviations in the manuscript such as CFL (L. 393) and GPR (L. 457).

Overall, major revision of the manuscript is recommended.

Author Response

The authors of the manuscript entitled "FDTD-Based Electromagnetic Modeling of Dielectric Materials with Fractional Dispersive Response" overview the fractional calculus models of dielectric response, propose the fractional polynomial-based dielectric function and thereupon derive fractional Maxwell's equations. As a welcome result, the calculations performed using the proposed FDTD scheme have been verified by accurate analytical calculations. Regretfully, the authors do not provide evidence of superiority of their approach over standard numerical solution of conventional Maxwell's equations. Neither do they support fractional calculations by experimental measurements. Together with the extreme difficulties, cumbersomeness, and computational demands in realizing fractional calculus in simulations, this casts doubts on practical usefulness of the approach proposed.

Answer

The Reviewer should note that the standard FDTD algorithm for the numerical solution of Maxwell’s equation is totally unable to handle dielectric materials with arbitrary dispersive response. Therefore, the fractional-calculus-based methodology illustrated in the presented article addresses and removes the aforementioned limitation, which is a major roadblock to transient EM field simulations in pulsed regime. The implementation of the proposed methodology is straightforward, and the relevant effectiveness and accuracy have been validated, successfully, by comparison with fully analytical results. The authors hope that this remark can help the Reviewer appreciate the contribution given to the scientific literature.

When reviewing literature on fractional calculus in dielectrics behavior, the authors causelessly ignore works of pioneers in the application of fractional calculus in the dielectric theory, Yu. Kalmykov and R. Hilfer. A number of recent studies of other authors within this area have been disregarded, too.

Answer

Many thank for the remark. In order to give positive feedback to the reviewer suggestion the references Kalmykov, Y.P.; Coffey, W.T.; Crothers, D.S.F.; Titov, S.V. Microscopic models for dielectric relaxation in disordered systems. Phys. Rev. E 2004, 70, 041103, and Hilfer, R. Analytical representations for relaxation functions of glasses. J. Non-Crist. Solids 2002, 70, pp. 122-126, have been added.

In the Applications section, it is not clear where Eqs. (73)-(77) have been derived from. Are the polynomials in these relationships arbitrarily chosen or adjusted to fit the spectra?

Answer

Thanks for the remark. To this aim, included in the section 4.2 the following sentences “The exponents and coefficients characterizing the polynomials involved in (73) were arbitrarily chosen”, as well as “Also in this case, the polynomials in the relationships (74)-(76) were arbitrarily chosen”. 

The language of the manuscript requires certain improvement, too. This especially concerns the Applications section, where several cases of inappropriate wording have been noticed. In addition, there are unexplained abbreviations in the manuscript such as CFL (L. 393) and GPR (L. 457).

Answer

The language was revised as well as the abbreviations CFL and GPR were included.

Round 2

Reviewer 1 Report

I accept the revised version

Author Response

We thanks the reviewer for his/her positive feedback

Reviewer 3 Report

In the Response, the authors have addressed my comments and suggestions, but I still do not feel the Response (and the revision) to be quite satisfactory.

The authors insist that their study is a review. However, from looking at the proportion of the reviewed and original material (50:50), the impression is that the main purpose of the manuscript was to put the authors' own results in a wrapper of review. Moreover, the scientific value of the authors' original material is all the more questionable given the arbitrariness in choosing coefficients and exponents in the model polynomial to fit the anlytical results.

Furthermore, although in the revision the authors have cited some pioneering works on the fractional calculus in dielectrics theory hinted at in my previous report, this research was not comprehended by the authors and mentioned just passingly and, in my opinion, not in the most appropriate section. In a review, the cited literature should be overviewed comprehensively.

The authors have neglected my suggestion of paying attention to recent works on fractional dielectric function. I recommend finding an appropriate context within the present review for the following studies:

D. Zhao, H. Sun (2019) Z. Angew. Math. Phys. 70, 42 (and probably, references therein);

works of K. Górska, A. Horzela et al.

Author Response

Following the Reviewer’s suggestion, the authors have included references to studies of D. Zhao, H. Sun (2019). The authors would like to thank the Reviewer for his/her suggestion that has resulted in an improvement of the paper.

Round 3

Reviewer 3 Report

This authors' revision can be recommended for publishing.